# Role of Hyaluronan in Human Adipogenesis: Evidence from in-Vitro and in-Vivo Studies

**DOI:** 10.3390/ijms20112675

**Published:** 2019-05-31

**Authors:** Nicholas Wilson, Robert Steadman, Ilaria Muller, Mohd Draman, D. Aled Rees, Peter Taylor, Colin M. Dayan, Marian Ludgate, Lei Zhang

**Affiliations:** 1School of Medicine, Cardiff University, Heath Park, Cardiff CF14 4XN, UK; nickmwilson1@gmail.com (N.W.); SteadmanR@cardiff.ac.uk (R.S.); MullerI4@cardiff.ac.uk (I.M.); ReesDA@cardiff.ac.uk (D.A.R.); TaylorPN@cardiff.ac.uk (P.T.); DayanCM@cardiff.ac.uk (C.M.D.); Ludgate@cardiff.ac.uk (M.L.); 2Faculty of Medicine, Universiti Sultan Zainal Abidin, Jalan Sultan Mahmud, Kuala Terengganu 20400, Malaysia; DramanYusofMS@cardiff.ac.uk

**Keywords:** mesenchymal stem cell, Adipogenesis, fat accumulation, Extracellular matrix, hyaluronan, BMI, 4-methylumbelliferone, PPARγ

## Abstract

Hyaluronan (HA), an extra-cellular matrix glycosaminoglycan, may play a role in mesenchymal stem cell differentiation to fat but results using murine models and cell lines are conflicting. Our previous data, illustrating decreased HA production during human adipogenesis, suggested an inhibitory role. We have investigated the role of HA in adipogenesis and fat accumulation using human primary subcutaneous preadipocyte/fibroblasts (PFs, *n* = 12) and subjects of varying body mass index (BMI). The impact of HA on peroxisome proliferator-activated receptor gamma (PPARγ) expression was analysed following siRNA knockdown or HA synthase (HAS)1 and HAS2 overexpression. PFs were cultured in complete or adipogenic medium (ADM) with/without 4-methylumbelliferone (4-MU = HA synthesis inhibitor). Adipogenesis was evaluated using oil red O (ORO), counting adipogenic foci, and measurement of a terminal differentiation marker. Modulating HA production by HAS2 knockdown or overexpression increased (16%, *p* < 0.04) or decreased (30%, *p* = 0.01) PPARγ transcripts respectively. The inhibition of HA by 4-MU significantly enhanced ADM-induced adipogenesis with 1.52 ± 0.18- (ORO), 4.09 ± 0.63- (foci) and 2.6 ± 0.21-(marker)-fold increases compared with the controls, also increased PPARγ protein expression (40%, (*p* < 0.04)). In human subjects, circulating HA correlated negatively with BMI and triglycerides (*r* = −0.396 (*p* = 0.002), *r* = −0.269 (*p* = 0.038), respectively), confirming an inhibitory role of HA in human adipogenesis. Thus, enhancing HA action may provide a therapeutic target in obesity.

## 1. Introduction

The extracellular matrix (ECM) plays a substantial role in adipogenesis during mesenchymal stem cell (MSC) differentiation into fat [1,2]. Recent studies showed that hyaluronan (HA), a vital component of the ECM, plays an important role in adipogenesis [3,4]. HA is a non-sulphated linear glycosaminoglycan polymer composed of repeating D-glucuronic acid and N-acetylglucosamine monosaccharides. There are three isoforms of HA synthases (HASs) with distinct enzymatic properties to generate functional HA fragments through biosynthesis and degradations [3,5], with HAS2 being the major responsible isoform in human MSC, preadipocytes/fibroblasts (PFs), derived from human subcutaneous adipose tissues [6]. HASs synthesize high-molecular-weight (HMW)-HAs ranging from 2 × 10^5^ Da to 2 × 10^6^ Da, which have been proposed to have anti-inflammatory functions [5,7]. HMW-HAs are further processed and broken down to low-molecular-weight (LMW)-HAs (<120 kDa). These smaller HA fragments have been proposed to play roles in many chronic diseases including diabetes, autoimmune diseases, inflammation and cancer [7,8,9,10]. Strategies to reduce LMW-HA production or action have therefore been undertaken in animal models and pre-clinical studies, including adoption of the Food and Drug Administration (FDA)-approved drug 4-MU, an inhibitor of HA biosynthesis [11,12,13,14]. 

Recent studies have established a close correlation between HA and insulin resistance/obesity in murine models [4]. However, the role of HA in adipogenesis is still largely unknown. Previous studies from different research groups reported conflicting findings, with decreased adipogenesis being shown in response to either the inhibition of HA (e.g., 4-MU) [15] or by introducing enzymatically digested HA fragments (~50 kDa) [16,17] in in-vitro 3T3L1 cells or in in-vivo mouse models. These studies showed that the inhibition of adipogenesis was accompanied by decreased levels of peroxisome proliferator-activated receptor gamma (PPARγ), a master transcriptional regulator of adipogenesis [18]. Adipogenesis is lineage-specific differentiation of MSC driven by a transcriptional cascade in which PPARγ expression is upregulated early in the process and so its transcripts are measured to indicate early–intermediate adipogenesis [17,18]. This contrasts with lipoprotein lipase (LPL), an enzyme involved in lipid uptake, which is used as a marker of terminal differentiation. 

Previously, we reported that HA production was reduced during adipogenesis in primary human adipose-derived stem cells, mediated via HAS2 [19]. However, there remain no studies reporting the functional relevance of HA in human adipogenesis. In the current study, we thus employed a well-established human primary preadipocyte/fibroblasts (PFs) model [6,19] and a series of the modulation of HA approaches to characterise the role of HA in human adipogenesis. We also extended our observations to explore the correlation of estimated circulating hyaluronan per adipose tissue with body mass in healthy, overweight or obese individuals [20].

## 2. Results

### 2.1. Subsection

#### 2.1.1. HA Affects PPARγ Expression in Human PFs 

We treated human subcutaneous PFs with siRNA to knock down HAS enzyme isoforms; we produced a 58% or 56% reduction in HAS1 or HAS2 transcript levels, respectively, as previous reported [6]. There was a significant 48% decrease or 16% increase in PPARγ transcript levels following HAS1 and HAS2 knock-down, respectively, compared with the control ‘scrambled’ siRNA (Figure 1A). Our previous study revealed that the knockdown of the HAS1 isoform could lead to a compensatory increase in HAS2 transcripts and HA production (data not shown) [6]. Therefore, to delineate whether this effect is compensatory or a true consequence of a reduction in HAS1 and HAS2 expression, we tested the effects of HAS1 and HAS2 overexpression and initially used retroviral vectors to introduce these genes into subcutaneous PFs; however, the cells proliferated poorly and were not studied further.

As an alternative, we generated HEK293 cell lines stably expressing HAS1 or HAS2 as a source of HA-specific isoform-enriched supernatants (10–20-fold increase in HA compared to the control cells). The treatment of non-modified subcutaneous PFs with HA-enriched supernatants from HAS2–HEK293 in a serum-free medium showed a significant 30% reduction in PPARγ transcripts; there were no differences observed with HAS1–HEK293 or the control HEK293 cells transfected with empty vector (Figure 1B). These data suggest that the HA generated contributes to a negative regulation of PPARγ expression and may play an inhibitory role in adipogenesis.

#### 2.1.2. Inhibiting HA Production Enhances Adipogenesis

Human subcutaneous PFs were cultured in ADM for 22 days when differentiated cells displayed cell rounding and lipid droplet formation, a 2- to 3-fold increase in PPARγ transcripts (marker of intermediate adipogenesis) and the induction of LPL expression (marker of late adipogenesis), all as previously reported [21]. To investigate the possible role of HA in regulating adipogenesis in subcutaneous PFs, we conducted further experiments using different concentrations of the HA biosynthesis inhibitor, 4-methylumbelliferone (4-MU) in conjunction with the adipogenic cocktail. The 4-MU treatment has a significant reduction in HA in the cell culture supernatant as expected (data not shown). Preliminary experiments revealed a concentration-dependent increase in induced adipogenesis with 0.1 mM 4-MU being optimum for differentiation (Appendix A). 

We then examined the effect of 0.1 mM 4-MU on in vitro-induced adipogenesis in further samples (*n* = 6) using morphological assessment, semi-quantitative ORO staining or QPCR measurement of transcripts for LPL. The results are summarised in Figure 2 and show that adipogenesis was significantly enhanced (1.5- to 4-fold) in PFs irrespective of the evaluation method. 

Furthermore, PPARγ transcript (Figure 3A) and nuclear protein (Figure 3B) expression, especially of the fat-specific isoform PPARγ2 inducible by adipogenesis [22], were significantly increased by the inhibition of HA synthesis in both complete culture medium or adipogenic medium. 

#### 2.1.3. Low-Molecular-Weight (LMW) HA Exerts an Inhibitory Effect upon Adipogenesis 

Various studies investigating the role of HA in cellular development have found that the influence of HA on many cellular processes is dependent on molecular weight. Subcutaneous PFs were thus cultured with different molecular weights of HA in adipogenic medium. The treatment of PFs with 41–65 kDa LMW-HA resulted in an inhibition of adipogenesis with a significant reduction in LPL expression (56.6 ± 19.8% reduction, *p* < 0.03) (Figure 4). This result was confirmed by ORO staining, which showed a reduction of 16.9 ± 9.3% (*p* < 0.02) compared to the untreated control cultures. In contrast, high-molecular-weight (HMW)-HA isoforms (151–300 kDa and 1.01–1.8 MDa) showed no effect on adipogenesis. 

#### 2.1.4. Correlation of BMI with Circulating HA Concentrations

The above findings suggested that HA has an inhibitory role in adipogenesis; therefore, we sought to establish whether there was a relationship between serum HA concentrations and adipose tissue accumulation in a clinical study undertaken in participants with varying body weight [20] (Appendix A). Across all 60 participants, serum HA ranged from 16–39 ng/mL but did not correlate with BMI or triglyceride levels. However, the correction of serum HA for the differences in indexed the blood volume (details in Method section) showed that estimated circulating HA per adipose tissue (ng/kg) correlated negatively and significantly with both BMI (*r* = −0.396) and triglyceride levels (*r* = −0.269) (Figure 5).

## 3. Discussion

Our study demonstrated an inhibitory role of HA in human adipogenesis. We used an MSC model, PFs, derived from human subcutaneous fat [23] rather than a murine cell line, with the benefit of studying primary cells through adipogenesis in a more relevant physiological form. This well-established cell model is from the entire stromal vascular fraction, which is a heterogeneous population of PFs at varying stages of commitment. 

HA is located extracellularly and is needed to form structures for the development of mature adipocytes; increasing evidence suggests a strong correlation between HA and adipogenesis [4]. Our current study suggests that HA produced from HAS2 plays a repressor role in the regulation of the key transcription factor of adipogenesis, PPARγ [18], as evidenced by our experiments using siRNA knockdown or exposure to supernatants enriched with HA. Other studies have also shown an important role of PPARγ in the HA regulation of adipogenesis, albeit using cell lines [15,16]. HA may interact with its receptor CD44 to trigger IGF1 signalling [24], which is essential in adipogenesis through PPARγ regulation [2,25]. Furthermore, the chemical inhibition of HA production using 4-MU significantly enhanced adipogenesis, which is accompanied by increased PPARγ expression, particularly the adipose-specific isoform PPARγ2 [22]. The detailed mechanisms of HA functionality in stem cells via different isoforms of HAS is still largely unknown. Our current study indicates that the reduced HAS2 transcripts and HA production through subcutaneous adipogenesis reported previously [19] may further support adipogenesis through PPARγ2 regulation.

Studies have suggested an important role of HA and adipogenesis in obesity, diabetes and inflammation, and indicated a negative influence of LMW-HA (<120 kDa) [4,7,8,13]. For example, LMW-HA fragments contribute to the release of pro-inflammatory cytokines [26] or the impact of HA fragments on adipogenesis via interacting with its receptors [27]. In the current study, primary human PFs overexpressing HAS1 and HAS2 showed impaired proliferation (data not shown), supporting the importance of HA in stem cell function [28]. Furthermore, our study provides evidence that the LMW-HA, 41–65 kDa fragments, can inhibit adipogenesis in human subcutaneous PFs but not the HMW-HA fragments of 151–300 kDa and 1.01–1.8 MDa. It is important to note these LMW-HA fragments (41–65 kDa) are generated from HMW-HA produced by HAS2, which is over 2 × 10^6^ Da [27]. Future investigations are needed to clarify the mechanisms of the metabolism of HA produced from HAS2 required to suppress adipogenesis. 

HA has been identified as a potential therapeutic candidate in a range of diseases. For example, the high viscoelastic properties of HMW-HA may support its use in visco-supplementation for joint arthritis and as a scaffold for tissue engineering, whereas the inhibition of LMW-HA may be effective in diabetes, inflammation and cancer [13,14]. In addition, HA and stem cells/adipocytes have both been used in the cosmetic industry due to their properties of absorbing water (1000× HA weight) or ability to differentiate to enlarged mature adipocytes, respectively [29,30,31]. It is vital, therefore, to clarify the role of HA in human adipogenesis in accordance with these therapeutic and industry needs. Many studies in the field previously used mouse cell lines or animal models, which may not reflect human physiology due to differences in metabolism or drug delivery system. To add to this complexity, different depots of human fat may display different properties with respect to the actions of HA in adipogenesis [19]. For example, in contrast to the reduced HA observed through adipogenesis in subcutaneous PFs, a positive association between HA production and adipogenesis was observed in human PFs derived from orbital adipose tissues [19]. Such depot-specific differences in adipose function are well recognised more generally in human adipogenesis, with relevance to obesity and metabolic risk [32]. Our study provides the first insight of an inhibitory role of a vital ECM component, HA, in human adipogenesis.

Our clinical study enrolled samples from female participants free from metabolic diseases, with the exception of intentional differences between groups with respect to BMI [20]. Whilst serum HA concentrations showed no correlation with BMI, additional adjustment for the well-established differences in blood volume in obesity [33] showed that the estimated circulating HA per adipose tissue correlated negatively and significantly with BMI. To reflect the true effects of local factors on adipose tissues in obesity, improved models need to be developed for human studies. 

In conclusion, we have confirmed an inhibitory role of HA in human adipogenesis and established a negative correlation of blood volume-adjusted HA concentrations with BMI and triglyceride levels. This may reflect an association of lower HA production with increased amounts of adipose tissue. This observation agrees with our previous finding of decreased HA production in human subcutaneous adipogenesis [19]. Furthermore, the enhanced in vitro adipogenesis by the inhibition of HA shown in our current study indicates that a naturally decreased HA production may act as a facilitator of adipogenesis.

## 4. Materials and Methods 

All reagents were obtained from Sigma-Aldrich (Dorset, UK) and tissue culture components from Cambrex (Thermo Fisher Scientific, Waltham, MA, USA) unless otherwise stated.

### 4.1. Adipose Tissue Collection and Preparation

Adipose Tissue was collected with informed consent and local research ethics committee approval (reference number 06/WSE03/37) from patients undergoing elective open abdominal surgery for non-metabolic conditions. Preadipocytes/fibroblasts (PFs) were obtained by collagenase digest, as previously described (13). PFs were cultured in DMEM/F12 with high glucose (17.5 mM) and 10% FCS (complete medium, CM) and were used at a low passage number (<5). Hence, not all samples were analysed in all experiments. In all cases, adipogenesis was induced in confluent cells by replacing CM with adipogenic medium (ADM) containing 10% FCS, biotin (33 µM), panthothenate (17 µM), T3 (1 nM), dexamethasone (100 nM), thiazolidinedione (1 µM) and insulin (500 nM) for 22 days, as previously described [21].

### 4.2. Knockdown of HAS1 and HAS2 using siRNA

PFs were plated in 60-mm dishes until 70% confluent. Twenty-four hours before transfection with siRNA, they were switched to CM without antibiotics. The cells were incubated at 37 °C overnight in 2.5 ml culture medium containing 5 µL Lipofectamine 2000 (Invitrogen, Thermo Fisher Scientific, Waltham, MA, USA) and 1 µL 100 µM HAS1, HAS2 or scrambled siRNA (Applied Biosciences, Thermo Fisher Scientific, Waltham, MA, USA) to a final concentration of 40 nM. At the end of incubation, cells were cultured in a serum-free medium for 24 h, and RNA was collected for subsequent QPCR analysis [6]. 

### 4.3. HAS1, HAS2 Overexpression

Retroviral vectors (RV) for HAS1 and HAS2 were prepared by subcloning the coding regions into the pLNSX vector and sequenced. Plasmids containing HAS1/HAS2 or empty pLNSX (EMP) vectors were used for subsequent transfection of ‘Phoenix’ retroviral packaging cell lines as previously described [34]. The RV supernatants were used to transduce subcutaneous precursors (PFs) and the HEK293 cell line. Geneticin selection produced mixed cell populations with stable incorporation of HAS1/HAS2 or EMP. HA-enriched supernatants from HEK293 cells were collected over 24 h in a serum-free medium (10–20-fold higher levels of HA were detected by ELISA in HEK293–HAS1 or HEK293–HAS2 supernatants compared with HEK293–EMP).

### 4.4. Effect of the Inhibition of HA Synthesis on Induced Adipogenesis

HA synthesis can be inhibited using the methylcoumarin derivative, 4-methylumbelliferone (4-MU). Experiments were performed in which confluent PFs were incubated in ADM, with or without 0.1, 0.2, 0.4 or 1 mM 4-MU present for the entire period of adipogenesis induction. Adipogenesis was assessed by microscopic examination to detect the characteristic morphological changes (cell rounding, accumulation of lipid droplets), acquisition of lipid-filled droplets (oil red O staining, ORO) and transcript measurement of the adipogenic terminal differentiation marker lipoprotein lipase (LPL) by QPCR as described previously [35]. In addition, foci of differentiation (groups of cells with lipid droplets) were counted in ten different fields for each experimental condition [36].

### 4.5. QPCR of PPARγ, LPL Genes

The various cell populations were plated in 6-well plates in CM. RNA was extracted and reverse transcribed using standard protocols [21] for QPCR (quantitative PCR) analysis; primers were designed using primer 3 software (Appendix A). QPCR was conducted using SYBR Green incorporation measured on a Stratagene MX 3000. A comparison with plasmid standard curves for each gene permitted the calculation of absolute values for each sample (transcripts/μg input RNA). In addition, transcripts for a housekeeping gene, APRT, were measured so that the values could be expressed relative to this (transcripts/1000 APRT). It should be noted that none of the treatments used resulted in a variation in the APRT Ct value of more than 1 cycle. In QPCR experiments, all measurements were made in triplicate; the standard curve was also run in at least duplicate. 

### 4.6. Western Blotting

Confluent subcutaneous PFs were cultured in CM or ADM for 22 days; cells were harvested in ice-cold PBS, centrifuged and resuspended in HEPES buffer containing protease inhibitors. The supernatant produced by centrifugation at 12,000× *g* provided the cytosolic fraction, and high salt extraction of the pellet to provide nuclear fraction. Nuclear protein samples were analysed by Western blotting using anti-PPARγ (from Cell Signalling Technology, Danvers, MA, USA) and anti-actin (from Santa Cruz Biotechnology, Dallas, Texas, USA) antibodies as previously described [25]. 

### 4.7. HA Treatment

Three different molecular weight HA fragments were chosen: 41–65 kDa, 151–300 kDa and 1.01–1.8 MDa (LifeCore Biomedical, LLC. MN, USA). For convenience, these will be referred to as 65 kDa, 300 kDa and 1 MDa respectively. The HA powder was dissolved in DMEM/F12 and further diluted in ADM to the desired concentration (100 and 200 μg/mL). On Day 0, cells were treated with HA fragments in ADM and replaced every 3 days until the end of adipogenesis.

### 4.8. Clinical Study Examining the Relationship between Serum HA Concentrations and BMI

Serum samples for the clinical study were selected from participants involved in the Controlled Antenatal Thyroid Screening (CATS) II study [20]. CATS II was a large multi-centred randomised controlled trial which investigated the long-term consequences of suboptimal gestational thyroid function. CATS II recruited 480 participants from the original CATS I cohort [37]. In the current study, samples were excluded from participants who had a history of abnormal thyroid function or metabolic disease, either during pregnancy (CATS I) or after 9 years follow-up. Samples were retrieved from participants with healthy (<25 kg/m^2^), overweight (25–30 kg/m^2^) or obese (>30 kg/m^2^) BMIs (*n* = 20 each; Appendix A). All serum samples were stored at −20 °C until measurement.

### 4.9. Measurement of HA

In experiments to measure HA, culture supernatants from PFs undergoing various treatments were centrifuged to remove dead cells and stored at −80°C until analysis. Serum and culture supernatant HA was measured using a commercially available enzyme-linked immunosorbent assay (Corgenix) according to the manufacturer’s instructions [6].

### 4.10. Blood Volume-Adjusted Circulating HA Concentrations

There are significant physiological differences between normal and obese patients. As a patient deviates from ideal body weight and becomes obese, their blood volume increases [38]. When this increase in blood volume (mL) is considered relative to the increase in body weight (kg), the indexed blood volume (mL/kg) falls in a non-linear manner [33,39]. Lemmens et al. found that the indexed blood volume per adipose tissue can be estimated from BMI using the following equation [33]:$$Indexed\;Blood\;Volume\;\left(ml/kg\right)=\frac{70}{\sqrt{BMI/22\;}}$$

In order to reflect the available serum HA for the disproportionate increase in adipose tissues in obese individuals, serum HA concentration was adjusted for the indexed blood volume using the following equation: Estimated circulating HA per adipose tissue (ng/kg) = Serum HA concentration (ng/ml) × Indexed blood volume (mL/kg).

### 4.11. Statistical Analysis

Results were analysed using SPSS version 27. In all cases, *p* < 0.05 was considered significant. Differences between groups were analysed using ANOVA. For the clinical study, characteristics were analysed for normality using the Kolmogorov–Smirnov test. Correlation between normally distributed data was analysed using Pearson’s correlation and by Spearman’s rank for non-parametric data. 

## Figures and Tables

**Figure 1 ijms-20-02675-f001:**
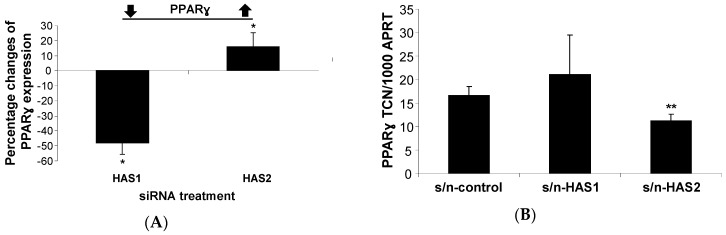
Hyaluronan (HA) from HA synthase (HAS)2 affects peroxisome proliferator-activated receptor gamma (PPARγ) transcripts. Subcutaneous preadipocytes/fibroblasts (PFs) (*n* = 4) were cultured until ~90% confluent, treated with HAS siRNA (**A**) or a hyaluronan-enriched supernatant (**B**) from HAS1 or HAS2 stable expressing HEK293 cell lines in a serum-free medium for 24 h, compared to scrambled siRNA or a supernatant from HEK293 cells with empty vector controls, respectively. PPARγ transcripts were measured by Q-PCR. Results are expressed as increased or decreased percentage changes in comparison to the controls (**A**) or transcript copy number (TCN) per 1000 copies of housekeeper gene (adenosine phosphoribosyl transferase, APRT) (**B**). Histograms = the mean ± SEM of all samples studied. * *p* < 0.04; ** *p* ≤ 0.01.

**Figure 2 ijms-20-02675-f002:**
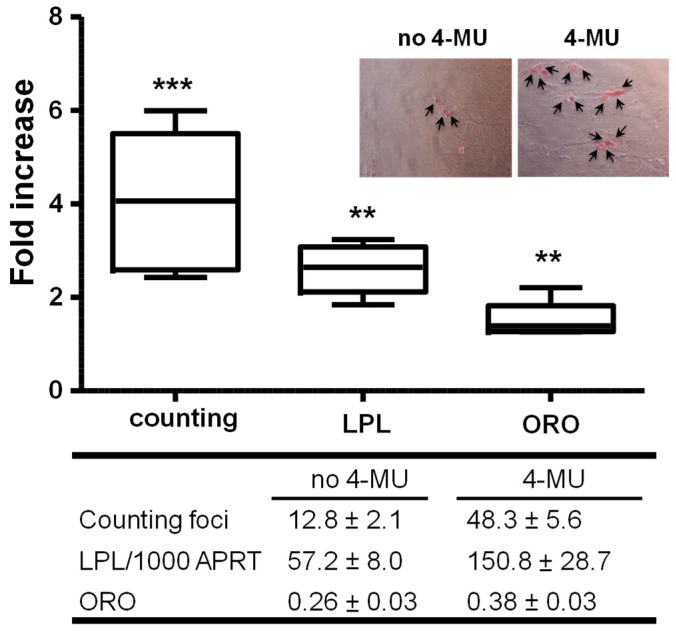
Enhanced adipogenesis by HA synthetic inhibitor, 4-MU. Confluent primary subcutaneous PFs were cultured in adipogenic medium (ADM) or complete medium (CM) with/without 4-MU for 22 days. Total RNA and nuclear protein were prepared. Fold effect (relative to the untreated control) of adipogenesis using foci counting (representative photos were shown with arrows indicating differentiating adipocytes), lipoprotein lipase (LPL) transcripts and oil red O (ORO) staining methods (*n* = 6). The table reports QPCR results together with foci numbers and ORO optical density values as the mean ± SEM. *p*-values are indicated in the figure above. ** *p* ≤ 0.01; *** *p* ≤ 0.006.

**Figure 3 ijms-20-02675-f003:**
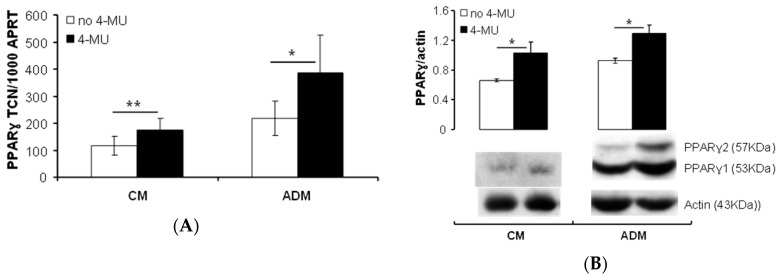
Enhanced PPARγ expression by 4-MU treatment. Confluent primary subcutaneous PFs were cultured in ADM or CM with/without 4-MU for 22 days. Total RNA and nuclear protein were prepared. (**A**) PPARγ transcripts (CM, *n* = 4; ADM, *n* = 8) and (**B**) nuclear protein (*n* = 4) were analysed by QPCR and Western blotting, respectively. Results are expressed as transcript copy number (TCN) per 1000 copies of housekeeper gene (adenosine phosphoribosyl transferase (APRT)) for QPCR, or PPARγ/actin for protein expression. Histograms = the mean ± SD (**A**) or SEM (**B**) of all samples studied. Two-tailed t-test has been used. * *p* < 0.04; ** *p* ≤ 0.01.

**Figure 4 ijms-20-02675-f004:**
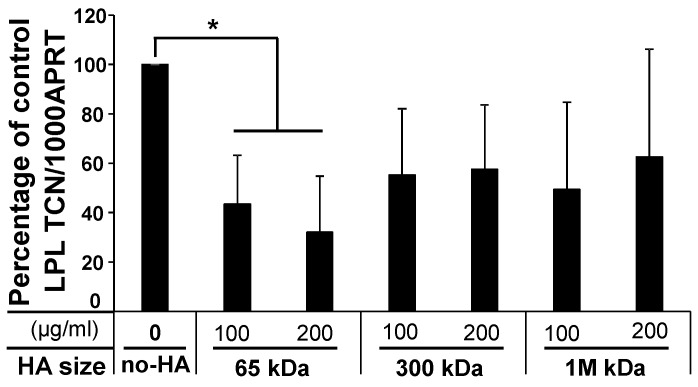
Decreased percentage of adiogenesis by LMW-HA treatment. LPL (terminal adipogenesis marker) was analysed by treatment of hyaluronan fragments (41–65 (65 kDa), 151–300 (300 kDa) and 1010–1800 (1 MDa)) in ADM compared to the no-HA control. Results are given as a percentage of the no-HA control (control as 100%) in LPL Transcript Copy Number (TCN)/1000 APRT. * *p* < 0.05.

**Figure 5 ijms-20-02675-f005:**
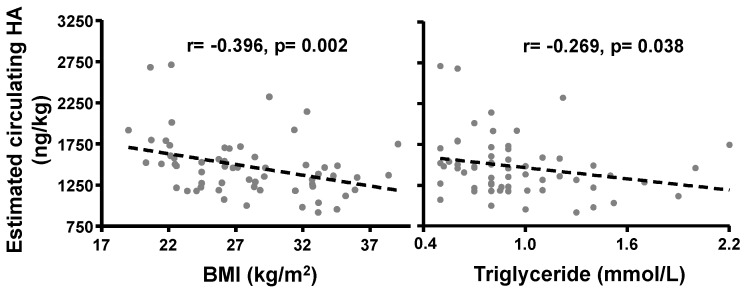
Negative correlations of estimated circulating HA per adipose tissue and BMI/triglyceride. The scatterplot showing the relationship between blood volume-adjusted circulating HA (ng/kg) and BMI (kg/m^2^), or triglyceride (mmol/L). Statistically significant Spearman’s correlations have shown between baseline characteristics and circulating HA.

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
