# Peer review of "Role of Hyaluronan in Human Adipogenesis: Evidence from in-Vitro and in-Vivo Studies"

_ijms, 2019, doi:10.3390/ijms20112675_

Round 1
Reviewer 1 Report
This manuscript addresses a very important topic, and the work will be of interest to a wide audience. There are a few important issues that need to be addressed, before the results can be considered to support the conclusions.
There are many conflicting reports on the signaling effect of low molecular weight HA. This reflects the fact that the levels of the relevant receptor or signal pathway proteins can vary. it is important to document the relevant initial cellular status, which may affect the result. For example, a recent report (Rotter Sopasakis et al. 2018; doi: 10.1177/1947603518770256) that the glucose concentration of cell culture medium can affect the expression of TLR4 in chondrocytes may be relevant to this study, since signaling by low molecular weight HA is often dependent on TLR4. The authors should specify the glucose concentration, and preferably TLR4 expression levels.
The calculation of blood volume-adjusted circulating HA appears to be erroneous in its basis, and improperly applied. A plot of the equation for indexed blood volume alone looks exactly like Figure 5 left. Multiplying a constant serum HA concentration in ng/ml by a decreasing function for indexed blood volume results in a decreasing function. This has no meaning. The conclusion that the quantity of circulating HA (but not its concentration) in the body correlates with BMI is not relevant.
Author Response
We would like to thank the reviewers for their insightful comments, for recognising that the studies have been well done and are of value for the understanding of the role of hyaluronan in human adipogenesis.
Rebuttal for ijms-505531 Wilson et al.
(Reviewers comments are in italics, while our response follows after each point)
Reviewer 1: Comments and Suggestions for Authors
This manuscript addresses a very important topic, and the work will be of interest to a wide audience. There are a few important issues that need to be addressed, before the results can be considered to support the conclusions.
There are many conflicting reports on the signaling effect of low molecular weight HA. This reflects the fact that the levels of the relevant receptor or signal pathway proteins can vary. it is important to document the relevant initial cellular status, which may affect the result. For example, a recent report (Rotter Sopasakis et al. 2018; doi: 10.1177/1947603518770256) that the glucose concentration of cell culture medium can affect the expression of TLR4 in chondrocytes may be relevant to this study, since signaling by low molecular weight HA is often dependent on TLR4. The authors should specify the glucose concentration, and preferably TLR4 expression levels.
We agree that it is important to clarify the cell culture condition, and stated in Line 225-226 “with high glucose (17.5mM) and…”. Our study has demonstrated that LMW-HA has negative impact on adipogenesis, we agree it would be interesting to have future investigations regarding TLR4 and LMW-HA in relating to adipogenesis processes.
We have cited the above study in the introduction section line 47 (reference 10).
The calculation of blood volume-adjusted circulating HA appears to be erroneous in its basis, and improperly applied. A plot of the equation for indexed blood volume alone looks exactly like Figure 5 left. Multiplying a constant serum HA concentration in ng/ml by a decreasing function for indexed blood volume results in a decreasing function. This has no meaning. The conclusion that the quantity of circulating HA (but not its concentration) in the body correlates with BMI is not relevant.
Our study has demonstrated an inhibitory role for HA on adipogenesis in vitro. To support this concept we attempted to evaluate circulating HA levels in obese individuals and compare it with those in subjects with lower BMI. Unfortunately this poses problems due to the relative decrease in blood volume as a proportion of the whole, as adipose tissue expands. Therefore, we used the adjusted indexed blood volume to calculate the estimated circulating HA per adipose tissues from a well-established model (Lemmens et al.).
However we agree this model may not be ideal, and have stated in the discussion, line 210-211 “To reflect the true effects of local factors on adipose tissues in obesity, improved models need to be developed for human studies.” We made changes using “estimated circulating HA per adipose tissue” accordingly: line 67-68; 123; 153, 209, 291, 296.
Reviewer 2 Report
The manuscript address a very important and not well understood issues such as the correlation between the extracellular matrix and the cell differentiation, moreover, since, as the authors suggest starting line 176, the use of both hyaluronan (HA) and adipocytes is increased in clinic and cosmetic uses, it is critical to deeply investigate their reciprocal effects.
Despite the interest in the subject and the authors knowledge of the issues, the manuscript presents some major flaws in the presentation of the data as well in the conclusion, resulting in some point difficult to follow and without firm ground.
In particular, I would suggest revising and possibly rewriting the abstract, adding more introductive aspects and less methodological data; the conclusive lines in the end seems not correlated to each other.
Line 47: e.g. USING 4-MU
Line 48: in IN VIVO mouse
Lines 46-50: it is not clear where are reported the “conflicting findings”?
Please, add in introduction some literature data about the adipogenesis process including PPAR-gamma and LPL that are introduced in the results but without any previous explanation (some readers may not be expert of adipogenesis) than lines 79-82 in the results section.
Lines 71-77: it should be best to add some data about the quantification and/or dimension of HA produced by modified HEK293, or at least move here the information reported in line 225.
Line 75-77: not clear why the author write of depleting HA since are commenting the addition of HA enriched media
line 86: 0.1 mM 4-MU is optimum for what? HA inhibition? Differentiation? >Please, explain and in case the authors refer to HA quantification, add the numbers. Change figure 1S with S1
figure 1: not easy to understand the negative change in siHAS1 treatment, better use a plot with a control reported to 100 and the samples decreased or increased. Moreover, in figure 4 are reported the changes or the percentage expression? It is also not clear since the control is reported as 100
For what it concerns figure 1, the authors only consider HAS1 and HAS2, although they correctly reported that there are 3 HASes; It is generally assumed that the most important HAS for HA synthesis is HAS2, but the authors should at least comment it and explain why they do not consider HAS3 at all. Can the author add a figure or comment the relative levels of expression of the 3 isoforms?
Figure 2: in the caption is reported the letter A) but not in the figure. The p value should be added in the table.
Figure 3: the authors used 2 different condition, ADM or CM for the cell culture, but they do not clearly explain why in the results as well as in conclusions. Figure 3 B in the CM samples is not analysed the isoform PPARgamma2, again without comments or explanation.
Figure 4: data are percentage changes or only percentages? Not clear for the presence of the 100% “no-HA” sample. The authors do not comment the correlation (if exist) with results in figure 1b. Why do they use 2 different markers (PPARgamma) and LPL? Are these markers interchangeable or do they have different meanings?
Figure 5: add measure units; line 138: Spearman
The authors do not explain clearly the meaning of the “negatively and significantly” correlation between circulating HA and BMI. The way this result is reported in the manuscript seems quite fortuitus, while it can be fundamental for the research and of great clinical relevance.
Line 150: eliminate “isoforms” unless the authors can indicate the molecular dimension of the HA in the conditioned media, with respect to the standards used in figure 4.
Line 155-157: this sentence can be moved in the introduction section.
Line 165: indicate that data about the proliferation are “data not shown”
Line 167 and following: comment the correlation (if any) with data in figure 2B
Line 173-176: I suggest eliminating this sentence in order to simplify the discussion, since the model is quite different.
line 236: 6-well plates or 60 mm Petri dishes?
Line 302: ANOVA; eliminate “consider”
Line 308: “effect”.
Author Response
We would like to thank the reviewers for their insightful comments, for recognising that the studies have been well done and are of value for the understanding of the role of hyaluronan in human adipogenesis.
Rebuttal for ijms-505531 Wilson et al.
(Reviewers comments are in italics, while our response follows after each point)
Reviewer 2: Comments and Suggestions for Authors
The manuscript address a very important and not well understood issues such as the correlation between the extracellular matrix and the cell differentiation, moreover, since, as the authors suggest starting line 176, the use of both hyaluronan (HA) and adipocytes is increased in clinic and cosmetic uses, it is critical to deeply investigate their reciprocal effects.
Despite the interest in the subject and the authors knowledge of the issues, the manuscript presents some major flaws in the presentation of the data as well in the conclusion, resulting in some point difficult to follow and without firm ground.
In particular, I would suggest revising and possibly rewriting the abstract, adding more introductive aspects and less methodological data; the conclusive lines in the end seems not correlated to each other.
We agree and prepared another abstract as below within the 200 words limit (line 12-30).
Hyaluronan (HA), an extra-cellular matrix glycosaminoglycan, may play a role in mesenchymal stem cell differentiation to fat but results using murine models and cell-lines are conflicting. Our previous data, illustrating decreased HA production during human adipogenesis, suggested an inhibitory role. We have investigated HA’s role in adipogenesis and fat accumulation using human primary subcutaneous preadipocyte/fibroblasts (PFs, n=12) and subjects of varying body mass index (BMI). The impact of HA on PPARγ expression was analysed following siRNA-knockdown or overexpression of HA synthase (HAS)1 and HAS2. PFs were cultured in complete or adipogenic medium (ADM) with/without 4-Methylumbelliferone (4MU= HA synthesis inhibitor). Adipogenesis was evaluated using oil-red-O (ORO), counting adipogenic foci, and measurement of a terminal differentiation marker. Modulating HA production by HAS2 knockdown or over-expression increased (16%, p<0.04) or decreased (30%, p=0.01) PPARγ transcripts respectively. HA inhibition by 4-MU significantly enhanced ADM-induced adipogenesis with 1.52±0.18 (ORO), 4.09±0.63 (foci) and 2.6±0.21 (marker) fold increases compared with controls and increased PPARγ protein (40%, (p<0.04)). In human subjects circulating HA correlated negatively with BMI and triglycerides (r = -0.396 (p=0.002), r = -0.269 (p=0.038), respectively), confirming an inhibitory role of HA in human adipogenesis. Thus enhancing HA action may provide a therapeutic target in obesity.
Line 47: e.g. USING 4-MU
Addressed line 49
Line 48: in IN VIVO mouse
Addressed line 55
Lines 46-50: it is not clear where are reported the “conflicting findings”?
Addressed line 53 and line 54 as: Previous studies “from different research groups” reported conflicting findings, with decreased adipogenesis being shown in response to “either” inhibition of HA (e.g. 4-MU) [15] or by introducing enzymatically digested HA fragments (~50 kDa) [16, 17] in in vitro 3T3L1 cells or in in vivo mouse models.”
Please, add in introduction some literature data about the adipogenesis process including PPAR-gamma and LPL that are introduced in the results but without any previous explanation (some readers may not be expert of adipogenesis) than lines 79-82 in the results section.
Line 58-61 “Adipogenesis is lineage-specific differentiation of MSC driven by a transcriptional cascade in which PPARγ expression is upregulated early in the process and so its transcripts are measured to indicate early-intermediate adipogenesis [17, 18]. This contrasts with lipoprotein lipase (LPL), an enzyme involved in lipid uptake, which is used as a marker of terminal differentiation.” has been added.
Lines 71-77: it should be best to add some data about the quantification and/or dimension of HA produced by modified HEK293, or at least move here the information reported in line 225.
Line 83: added “(10-20 fold increased HA compared to control cells)”
Line 75-77: not clear why the author write of depleting HA since are commenting the addition of HA enriched media
Changed to line 87-88 “These data suggest that the HA generated contributes to a negative regulation of PPARγ expression and may play an inhibitory role in adipogenesis”
line 86: 0.1 mM 4-MU is optimum for what? HA inhibition? Differentiation? >Please, explain and in case the authors refer to HA quantification, add the numbers. Change figure 1S with S1
line 100 added “for differentiation”, also changed 1S to S1.
figure 1: not easy to understand the negative change in siHAS1 treatment, better use a plot with a control reported to 100 and the samples decreased or increased.
We have made changes of the legend as “Results are expressed as increased or decreased percentage changes in comparing to controls (A) (line 131)”. We have made a new figure 1A as below:
Moreover, in figure 4 are reported the changes or the percentage expression? It is also not clear since the control is reported as 100
Addressed for the same comments as below.
For what it concerns figure 1, the authors only consider HAS1 and HAS2, although they correctly reported that there are 3 HASes; It is generally assumed that the most important HAS for HA synthesis is HAS2, but the authors should at least comment it and explain why they do not consider HAS3 at all. Can the author add a figure or comment the relative levels of expression of the 3 isoforms?
We added Line 39-42 as comments below to address the importance of HAS2 and HA in human PFs in our study, therefore we did not consider to investigate HAS3 isoform. “There are three isoforms of HA synthases (HAS) with distinct enzymatic properties to generate functional HA fragments through biosynthesis and degradations [3, 5], with HAS2 being the major responsible isoform in human MSC, preadipocytes/fibroblasts (PFs), derived from human subcutaneous adipose tissues [6].”
Since we have previously published the compensate effect of HAS2 with QPCR measurements, therefore we did not show this data in current study as stated in line 77 “(data not shown)”.
Figure 2: in the caption is reported the letter A) but not in the figure. The p value should be added in the table.
Deleted “A)” (line 137); line 140-141 added “p values indicated in the figure above”
Figure 3: the authors used 2 different condition, ADM or CM for the cell culture, but they do not clearly explain why in the results as well as in conclusions. Figure 3 B in the CM samples is not analysed the isoform PPARgamma2, again without comments or explanation.
In line 106 as: Furthermore, the expression of PPARγ transcripts (figure 3A) and nuclear protein (figure 3B), especially of the fat-specific isoform PPARγ2 “inducible by adipogenesis” [22].
Figure 4: data are percentage changes or only percentages? Not clear for the presence of the 100% “no-HA” sample. The authors do not comment the correlation (if exist) with results in figure 1b. Why do they use 2 different markers (PPARgamma) and LPL? Are these markers interchangeable or do they have different meanings?
Line 149-152 changed to “Decreased percentage of adiogenesis by LMW-HA treatment.” and “Results are given as percentage of the no-HA control (control as 100%) in LPL Transcript Copy Number (TCN)/1000 APRT.”
Line 58-61 “Adipogenesis is lineage-specific differentiation of MSC driven by a transcriptional cascade in which PPARγ expression is upregulated early in the process and so its transcripts are measured to indicate early-intermediate adipogenesis [17, 18]. This contrasts with lipoprotein lipase (LPL), an enzyme involved in lipid uptake, which is used as a marker of terminal differentiation.” has been added.
Figure 5: add measure units; line 138: Spearman
Added measure units and addressed line 155 “Spearman”.
The authors do not explain clearly the meaning of the “negatively and significantly” correlation between circulating HA and BMI. The way this result is reported in the manuscript seems quite fortuitus, while it can be fundamental for the research and of great clinical relevance.
As replied to review 1: Our study has demonstrated an inhibitory role for HA on adipogenesis in vitro. To support this concept we attempted to evaluate circulating HA levels in obese individuals and compare it with those in subjects with lower BMI. Unfortunately this poses problems due to the relative decrease in blood volume as a proportion of the whole, as adipose tissue expands. Therefore, we used the adjusted indexed blood volume to calculate the estimated circulating HA per adipose tissues from a well-established model (Lemmens et al.).
However we agree this model may not be ideal, and have stated in the discussion, line 210-211 “To reflect the true effects of local factors on adipose tissues in obesity, improved models need to be developed for human studies.” We made changes using “estimated circulating HA per adipose tissue” accordingly: line 67-68; 123; 153, 209, 291, 296.
Line 150: eliminate “isoforms” unless the authors can indicate the molecular dimension of the HA in the conditioned media, with respect to the standards used in figure 4.
Deleted “isoforms” in line 167
Line 155-157: this sentence can be moved in the introduction section.
Agree, moved to Line 39-43 as “There are three isoforms of HA synthases (HAS) with distinct enzymatic properties to generate functional HA fragments through biosynthesis and degradations [3, 5], with HAS2 being the major responsible isoform in human MSC, preadipocytes/fibroblasts (PFs), derived from human subcutaneous adipose tissues [6]. HAS synthesize high molecular weight (HMW)-HAs ranging from 2 x 105 Da to 2 x 106 Da, which have been proposed to have anti-inflammatory functions [5, 7].”
Line 165: indicate that data about the proliferation are “data not shown”
Added “data not shown” in line 182
Line 167 and following: comment the correlation (if any) with data in figure 2B
We don’t have figure 2B, have deleted “A)” from legend of figure 2 (line 137).
Line 173-176: I suggest eliminating this sentence in order to simplify the discussion, since the model is quite different.
Line 185-186, deleted sentence as suggested, “Such findings are consistent with observations reported previously in murine 3T3-L1 cells”.
line 236: 6-well plates or 60 mm Petri dishes?
Adjusted line 254, 6-well plates.
Line 302: ANOVA; eliminate “consider”
Adjusted line 320
Line 308: “effect”.
Adjusted line 326
Reviewer 3 Report
In the present publication, the authors investigated the impact of hyaluronan (HA) on the adipogenic differentiation capacity of human preadipocytes in vitro. siRNA knockdown of HAS2 increased the expression levels of the adipogenic marker PPARγ, whereas the treatment of the cells with supernatant of HAS2-overexpressing HEK293 cells decreased the expression levels of PPARγ. The adipogenesic differentiation is improved in presence of the HA synthesis inhibitor 4-Methylumbelliferone. The in vitro data are supported by some in vivo data showing a negatively correlation of the circulating hyaluronan with the BMI and triglycerides in patients.
Altogether the experiments are well-done and well presented. However for the publication in International Journal of Molecular Science, I suggest some additional experiments and improvements.
Although the term “mesenchymal stem cells” is controversial discussed in the literature, the authors should term their cells consistently as preadipocyte fibroblasts (PFs) or mesenchymal stem cells (MSCs). In the discussion part the authors call their cells MSCs. Prior to that they named them PFs
The authors mention that the knockdown of HAS1 could be compensated by HAS2 which is known to be the highest expressed HAS isoform. Here, the authors should perform an expression analysis of the three known HAS isoforms by qPCR. Also the inhibition of the HA synthesis by 4-MU should be confirmed by qPCR or by ELISA. Since HAS1 and HAS2 form different amounts of HA, the HA content in the supernatants of the HAS1- and respectively HAS2-overexpressing HEK293 cells should be determined.
The statistical analysis should be controlled. The authors should show the standard deviation, especially when the number of done experiments is different (Fig. 3a). The student T test must be used for the comparison of the means of two groups.
Author Response
We would like to thank the reviewers for their insightful comments, for recognising that the studies have been well done and are of value for the understanding of the role of hyaluronan in human adipogenesis.
Rebuttal for ijms-505531 Wilson et al.
(Reviewers comments are in italics, while our response follows after each point)
Reviewer 3: Comments and Suggestions for Authors
In the present publication, the authors investigated the impact of hyaluronan (HA) on the adipogenic differentiation capacity of human preadipocytes in vitro. siRNA knockdown of HAS2 increased the expression levels of the adipogenic marker PPARγ, whereas the treatment of the cells with supernatant of HAS2-overexpressing HEK293 cells decreased the expression levels of PPARγ. The adipogenesic differentiation is improved in presence of the HA synthesis inhibitor 4-Methylumbelliferone. The in vitro data are supported by some in vivo data showing a negatively correlation of the circulating hyaluronan with the BMI and triglycerides in patients.
Altogether the experiments are well-done and well presented. However for the publication in International Journal of Molecular Science, I suggest some additional experiments and improvements.
Although the term “mesenchymal stem cells” is controversial discussed in the literature, the authors should term their cells consistently as preadipocyte fibroblasts (PFs) or mesenchymal stem cells (MSCs). In the discussion part the authors call their cells MSCs. Prior to that they named them PFs
Line 159 added “PFs”, and made change to “PFs” accordingly, line 162; 182; 185; 201; 202
The authors mention that the knockdown of HAS1 could be compensated by HAS2 which is known to be the highest expressed HAS isoform. Here, the authors should perform an expression analysis of the three known HAS isoforms by qPCR.
Since we have previously published the compensate effect of HAS2 with QPCR measurements, therefore we did not show this data in current study as stated in line 77 “(data not shown)”.
Also the inhibition of the HA synthesis by 4-MU should be confirmed by qPCR or by ELISA.
In line 97-99 added “4-MU treatment has a significant reduction of HA in the cell culture supernatant as expected (data not shown).”
Since HAS1 and HAS2 form different amounts of HA, the HA content in the supernatants of the HAS1- and respectively HAS2-overexpressing HEK293 cells should be determined.
Line 83, added HA-specific isoform-enriched supernatants “(10-20 fold increased HA compared to control cells)”.
The statistical analysis should be controlled. The authors should show the standard deviation, especially when the number of done experiments is different (Fig. 3a). The student T test must be used for the comparison of the means of two groups.
We have added that two-tailed t-test has been used in the legend of figure 3. (line 147-148), and specified that mean±SD (A) or SEM (B) of all samples studied.
Round 2
Reviewer 2 Report
The authors replied to all comments raised in the first version of the manuscript.
Reviewer 3 Report
After the improvements the manuscript is well-done and well presented. Therefore I recommend to accept it in the present form.